# Draft Sequencing Crested Wheatgrass Chromosomes Identified Evolutionary Structural Changes and Genes and Facilitated the Development of SSR Markers

**DOI:** 10.3390/ijms23063191

**Published:** 2022-03-16

**Authors:** Jana Zwyrtková, Nicolas Blavet, Alžběta Doležalová, Petr Cápal, Mahmoud Said, István Molnár, Jan Vrána, Jaroslav Doležel, Eva Hřibová

**Affiliations:** Institute of Experimental Botany of the Czech Academy of Sciences, Centre of the Region Haná for Biotechnological and Agricultural Research, CZ-77900 Olomouc, Czech Republic; zwyrtkova@ueb.cas.cz (J.Z.); blavet@ueb.cas.cz (N.B.); dolezalova@ueb.cas.cz (A.D.); capal@ueb.cas.cz (P.C.); said@ueb.cas.cz (M.S.); molnar@ueb.cas.cz (I.M.); vrana@ueb.cas.cz (J.V.); dolezel@ueb.cas.cz (J.D.)

**Keywords:** *Agropyron cristatum*, annotation, chromosome sorting, chromosome-specific sequences, Illumina sequencing, SSR-marker development

## Abstract

Crested wheatgrass (*Agropyron cristatum*), a wild relative of wheat, is an attractive source of genes and alleles for their improvement. Its wider use is hampered by limited knowledge of its complex genome. In this work, individual chromosomes were purified by flow sorting, and DNA shotgun sequencing was performed. The annotation of chromosome-specific sequences characterized the DNA-repeat content and led to the identification of genic sequences. Among them, genic sequences homologous to genes conferring plant disease resistance and involved in plant tolerance to biotic and abiotic stress were identified. Genes belonging to the important groups for breeders involved in different functional categories were found. The analysis of the DNA-repeat content identified a new LTR element, *Agrocen*, which is enriched in centromeric regions. The colocalization of the element with the centromeric histone H3 variant CENH3 suggested its functional role in the grass centromere. Finally, 159 polymorphic simple-sequence-repeat (SSR) markers were identified, with 72 of them being chromosome- or chromosome-arm-specific, 16 mapping to more than one chromosome, and 71 mapping to all the *Agropyron* chromosomes. The markers were used to characterize orthologous relationships between *A. cristatum* and common wheat that will facilitate the introgression breeding of wheat using *A. cristatum*.

## 1. Introduction

Crested wheatgrass (*Agropyron cristatum* (L.) Gaertn.) belongs to the family Poaceae and the tribe Triticeae. It is a perennial, low-maintenance grass mainly used as forage and is grown worldwide [1,2]. Wheatgrasses are resistant to drought and cold [3,4,5,6], tolerant to environmental stress [7,8,9] and cadmium [10], and moderately tolerant to salinity. Wheatgrasses exhibit large genetic variation, which, together with their resistance to drought and biotic and abiotic stress, makes them a potential source of new genes for crop improvement [7,11,12]. Most *Agropyron* species are autotetraploids (2*n* = 4*x* = 28, PPPP) and can be found across Europe and Asia. Diploids (2*n* = 2*x* = 14, PP) are located on the same continents, excluding Turkey, Iran, and Georgia. Hexaploid species (2*n* = 6*x* = 42, PPPPPP) are less common and are predominantly located in Turkey, Iran, and Georgia [8]. Several artificial hybrids between common wheat (*Triticum aestivum*) and *A. cristatum* have been developed [13,14], and the introgression lines [15,16,17,18] have been used to improve wheat, for example, to increase the number of florets and kernels and enhance the grain weight and spike length [16,19].

The development of alien introgression lines, which involves hybridization and backcrossing with parental lines, is an important step in transferring useful genes and alleles from wild relatives to wheat [20]. A number of wheat–*Agropyron* chromosome addition and translocation lines have been developed and are used in wheat pre-breeding programs [17,21,22,23,24,25]. Wheat–*Agropyron* translocation lines bearing leaf-rust resistance, chromosomal loci in *A. cristatum*, conferring resistance to powdery mildew, and *Agropyron* chromosomes responsible for increased numbers of florets and kernels were recently identified [17,19,26]. The introgressions of genes of interest from wild relatives to important crops are hampered by linkage drag and extra chromatin parts that are transferred with the main trait loci [27]. Thus, the availability of molecular tools specific for *Agropyron* that can be used for the rapid characterization of breeding progeny is essential.

Several types of molecular markers originally developed for wheat, such as EST, STS, and COS markers, have also been applied for the characterization of *A. cristatum* [28,29,30]. Simple-sequence repeats (SSRs), or microsatellites, are a common and informative type of molecular marker, with high reproducibility and transferability and polymorphic patterns. They are one of the most informative genetic markers [31] used for the integration of genetic and physical maps [32] and for the ordering of genome scaffolds to larger genome-scale assemblies [33]. SSR markers have been widely used in association studies [34] and to analyze the genetic variability of natural populations of different plant species [34,35,36,37,38], as well as for fingerprinting and determining the genetic similarity of breeding lines [39]. One of the greatest advantages of SSR markers is their co-dominant character and their simple use as PCR-based markers.

Given the large genome size of *A. cristatum* (1C~6.3 Gbp) [40], its dissection into individual chromosomes by flow sorting could simplify genomic analyses [41]. This approach has facilitated the draft genome sequencing of barley, rye, wheat, chickpeas, and peas [42,43,44,45,46]. Sequencing data obtained from individual chromosomes or chromosomal arms can be used to identify genes of interest and to analyze genome structure through the creation of draft assemblies [43,44,47]. Chromosome-specific sequencing data were also used to develop different types of molecular markers and to map agronomic traits of interest [43,48,49,50]. However, obtaining chromosome-specific sequences may be hampered by difficulties in discriminating and sorting the chromosomes of interest. To overcome this obstacle, wheat-alien chromosome-addition lines have been used to separate individual chromosomes or chromosomal arms [19,25]. Alternatively, single chromosomes have been sorted from lines with standard karyotypes, followed by single-chromosome amplification [51].

In this work, we report on developing new genomic sources for *A. cristatum*. These include molecular markers to provide insights into the genome organization of *A. cristatum* and facilitate the introgression breeding of wheat.

## 2. Results

### 2.1. Chromosome Sorting and the Creation of Partial Assemblies

*A. cristatum* chromosomes 2P, 3P, 5P, and 6P were flow-sorted from bread wheat–*A. cristatum* addition lines at high purities, ranging from 81% (6P) to 98% (3P, 5P) (Table 1, Appendix A) and then amplified by whole-genome amplification [52]. Chromosomes 1P and 4P, which could not be discriminated from bivariate flow karyotypes (Appendix A), were sorted from cytogenetic stocks in a single-chromosome mode and used for single-chromosome DNA amplification [51]. Chromosome 7P was flow-sorted in a single-chromosome mode from a composite peak representing chromosomes 7P and 2P of the diploid genotype of *A. cristatum* cv. Parkway. Single-chromosome amplification [51] was followed by the identification of fractions of chromosomes 1P, 4P, and 7P. The first two chromosomes were identified using PCR markers targeting *Agropyron*-specific retroelements and chromosome 7P by PCR markers developed from cDNA sequences [40]. Each chromosome was amplified in at least four independent reactions to avoid DNA-amplification bias. Illumina sequencing resulted in 23.5×–42.8× coverage by the sequencing data, which were assembled separately for individual chromosomes. As expected, the quality of the partial chromosome assemblies differed between individual chromosomes (Table 1). Chromosomes 1P, 4P, and 7P, the sequencing libraries of which were prepared from DNA amplified in single-chromosome mode [51], resulted in assemblies of lower quality and represented only 4.3% to 15.4% of whole chromosome lengths according to Said et al. [40] (Table 1). The assemblies specific to chromosomes 2P, 3P, 5P, and 6P, which were sorted and amplified according to Šimková et al. [52], represented a higher percentage of their chromosome lengths (Table 1).

### 2.2. Genic Sequences Identified in the Assemblies

Partial chromosome-specific contigs and scaffolds enabled the identification of sequences homologous to important genes. Using the MAKER-P pipeline [53], we identified 45,126 coding and noncoding sequences and 44,814 coding/genic sequences (Table 1). The highest representation of genic sequences was found for chromosome 4P (~12%), and the lowest for chromosome 7P (~3%) (Appendix A), whereas the number of genic sequences per chromosome ranged from 1470 in chromosome 4P to 11,694 in chromosome 5P. The functional annotation provided a putative function for about 15% of the total number of these sequences. Of 20,616 gene families, most (15,877) were connected to proteins of unknown function. From the remaining 4,739 gene families with known function, two families were the most frequent: a family of kinase proteins, which included the rust-resistance kinases Lr10, and a cluster of germin-like proteins were found to be most abundant, especially in the genomic data for chromosome 3P (Figure 1). The most abundant genic sequences with a predicted function, which were annotated for all chromosomes, are summarized in Appendix A. Genes connected to forms of biotic resistance, such as the rust-resistance kinase *Lr10* (14 copies) and disease-resistance protein Piks-2 (three copies), were the most frequent on chromosome 3P, and the multidrug-resistance protein (four copies) was the most frequent on 4P. On the other hand, only a single copy of genes for tolerance, such as metal-tolerance protein C4 and 1, were found on chromosome 2P, and metal tolerance protein 7 and the chloroplast enhancing stress tolerance protein were found on chromosome 3P. Finally, genes increasing the fitness of the plant, such as abscisic stress-ripening protein, were found on chromosomes 2P, 7P (each with a single copy), and 3P (eight copies); heat-stress transcription factors were found on 2P, 3P, 6P (each with two copies), and 5P (seven copies); the zinc-finger-domain-containing stress-associated protein was found on 2P (single copy) and 5P (two copies); the salt stress root protein was found on 3P (single copy); transcription factors of the WRKY family were found on chromosome 3P (four copies) and 5P (three copies); and between two and ten copies of the transcription factors of the MYB family were found on each chromosome except for 4P (Appendix A).

Despite the low quality of the chromosome assemblies, sequences homologous to genes encoding agronomically important traits that can be utilized for wheat improvement were identified. Almost a complete sequence of the β-glucan-biosynthesis genes *HvCslF10* and *HvCslF3* was identified in *A. cristatum* data specific to chromosomes 2P and 3P; the arabinoxylan biosynthesis gene *TaGT47-13* was identified on 3P; and the bread-making-quality genes puroindoline A and B were identified on chromosome 5P. Genic sequences homologous to the leaf-rust-resistance gene *Lr1* were identified in the data for chromosome 5P, and the salt-tolerance genes *BP2A*, *SOS1*, and *HKT1* were found on chromosomes 3P and 7P. Sequences homologous to the vernalization locus *VRN2* were identified on chromosome 2P.

On chromosomes 2P and 5P, 5S rRNA genic sequences were found, whereas 45S rRNA genic sequences were found on chromosomes 1P, 4P, and 5P. The average representation of genic sequences on individual chromosomes was about ~5% of the assembly length.

### 2.3. Repetitive DNA Sequences Identified in the Assemblies

Repetitive DNA accounted for ~80% of the assemblies, excluding specific sequences, of 1P, which contained ~65% of DNA repeats, and 4P, with ~40% of repeats. To annotate repeats in the assemblies, public repeat databases (GenBank and Repbase), as well as the database of repeats identified and characterized in *A. cristatum* in our previous study [40], were used. As expected, DNA repeats were represented mainly by different types of transposable elements, and out of them, the *Athila* element of the Ty3/gypsy family was the most abundant (Appendix A). The proportion of individual types of transposable elements, as well as tandem organized repeats in chromosome assemblies, was similar to that in our previous study on the global analysis of repetitive DNA sequences identified in partial genomic Illumina sequence reads of *A. cristatum* cv. Parkway [40].

Our previous study revealed the presence of a relatively high number of tandem organized repeats, which were used as probes for FISH to identify wheatgrass chromosomes in situ [41]. Given that none of the tandem repeats were specifically localized to the centromeric regions of wheatgrass chromosomes, we speculated that centromeres of wheatgrass were enriched for specific LTR elements, similar to other related species in the Poaceae family [54]. Thus, we performed homology searches and phylogenetic analysis of *Cereba*-like elements, the most probable candidate of centromere-specific retroelements in Gramineae [54,55,56,57]. Complete sequences homologous to *Cereba*-like elements were identified in many scaffolds of all the chromosomes of *Agropyron* and were called *Agrocen* LTR elements. A phylogenetic analysis of the RT domain of *Agrocen* LTR elements confirmed their close relationship to centromeric retrotransposons (Appendix A; [57]). To confirm their location in the *A. cristatum* genome*,* FISH with probes specific to the RT domain of the most abundant *Agrocen* element was carried out. This work confirmed the presence of the *Agrocen* element in the centromeric regions of all the *Agropyron* chromosomes. Moreover, a combination of FISH and immunostaining with the grass centromere-specific histone H3 (CENH3; [58]) resulted in strong, overlapping signals (Figure 2).

### 2.4. Development of SSR Markers

The MISA program identified more than 134,000 SSRs with at least a 12-bp-long repetitive region containing 2- to 6-nt repetitive units. A majority of the SSRs were identified in the assemblies of chromosomes 2P, 3P, 5P, and 6P. The lowest number of SSRs (1136) was identified in the assembly of chromosome 4P, which represents only ~4% of its chromosome length (Table 1).

To increase the probability of identifying unique chromosome-specific markers, *Agropyron* scaffolds containing SSRs were aligned to the reference genome sequence of *T. aestivum* [46]. Scaffolds of *Agropyron* that mapped to a unique position in the wheat D-subgenome were then used to identify chromosome-specific SSRs. These SSRs were selected based on their positions in different locations along collinear chromosomes of the wheat D-subgenome and further verified by PCR on a set of wheat–*Agropyron* addition lines. Using this approach, we selected sets of 29 to 50 SSRs specific for each *Agropyron* chromosome (Appendix A).

### 2.5. Experimental Verification of Newly Designed SSR Markers

In total, 250 SSR primer pairs were selected for experimental verification. Out of them, 159 were found to be specific to *Agropyron*, including 16 that showed a polymorphic pattern (different lengths of the amplified PCR products) between *Agropyron* and bread wheat. A total of 72 SSR markers were specific for only one *Agropyron* chromosome, and 53 were specific for the chromosomal arm. There were 16 SSR markers that mapped to more than one chromosome or chromosomal arm and 71 SSRs that mapped to all the *Agropyron* chromosomes. Newly developed SSR markers were marked as “olomouc crested wheatgrass markers” (ocwgm).

The use of wheat chromosome-addition lines containing *Agropyron* chromosomal arms enabled us to assign some of the new SSR markers more precisely (Table 2, Figure 3). We identified six new SSR markers specific for chromosome 1P, with one of them unambiguously mapping on TH4 (1PS + 1BL) addition lines containing the short arm of chromosome 1P. In total, 13 SSRs mapped to chromosome 2P; out of them, 6 SSRs also mapped to the 2PS addition line and 6 other SSRs mapped to the 2PL addition line. The marker ocwgm019, which was found to be specific to chromosome 2P, also gave an amplification product on both the 2PS and 2PL cytogenetic lines. Out of the 14 SSRs specific to chromosome 3P, 9 were mapped to the 3PS addition line. Eleven SSR markers were found to be specific for chromosome 4P, with five of them also localizing to the 4PS addition line. Only four new SSR markers were found to be specific to chromosome 5P, with three of them mapping to the 5PL addition line. Out of the 15 SSRs specific to chromosome 6P, 8 were found on 6PS and 3 others on 6PL addition lines. Four markers that were unambiguously mapped to the addition line of chromosome 6P were not amplified on cytogenetic lines of 6PS or on 6PL. Finally, nine new SSR markers were found to be specific to chromosome 7P (the results are listed in Appendix A). As the addition line of chromosome 7P was not available to us, 7P-specific SSR markers were considered those that gave a specific product on *Agropyron* genomic DNA and simultaneously did not provide any positive PCR product on wheat–*Agropyron* addition lines. Given the fact that the wheat–*Agropyron* addition lines were created from different *Agropyron* cultivars, which could be genetically variable, we cannot exclude the possibility that some of the newly developed SSR markers could be absent on chromosome 7P.

Fourteen SSR markers were mapped to the addition line 3P + 3PS (the addition line including chromosome 3P and the extra short chromosomal arm 3PS), and out of them, five SSRs were missing in the addition line specific to 3PS; therefore, these SSR markers are most probably localized on the long chromosomal arm of 3P, the addition line of which was unavailable. A similar situation occurred during the analyses of chromosomes 4P and 5P. Six SSRs provided amplification products only in the addition line for chromosome 4P and simultaneously did not provide any amplification products in the addition line of 4PS; therefore, they were assigned to 4PL. One marker that only gave an amplification product in the addition line for chromosome 5P and not for 5PL was assigned to 5PS (Appendix A).

### 2.6. Orthologous Relationships between A. cristatum and Bread Wheat Chromosomes

Out of 72 (98.6%) single loci SSR markers, 71 showed synteny between bread wheat and the diploid *A. cristatum* genome (Figure 3, Appendix A). Only one marker (ocwgm059) identified in the scaffold of *A. cristatum,* which shared homology with the genomic sequence of chromosome group 1 of wheat, was mapped onto the nonhomologous chromosome 6PL in crested wheatgrass (Figure 3). Three SSR markers specific to chromosome 2P (ocwgm007, ocwgm011, and ocwgm013) were located in opposite chromosomal arms compared to wheat subgenomes, which indicated intrachromosomal rearrangement in *Agropyron* compared to wheat (Figure 4A). A similar situation was observed for 11 SSRs specific to chromosome 4P (ocwgm034–ocwgm044). Their position on short and long arms indicated the same structural rearrangement as observed for wheat chromosome 4A, which has an inverted structure in comparison to wheat chromosomes 4B and 4D (Figure 4B).

Besides the SSRs pointing to intrachromosomal changes, 16 additional SSR markers found at single loci in the wheat genome were detected as duplications in the genome of the diploid wheatgrass cultivar Parkway (Appendix A). For example, three markers specific to chromosome group 6 in wheat showed the duplication of 2P/6P, two markers showed the duplication of 3P/4P, and two SSRs indicated the duplication of 1P/5P/6P. Other duplicated regions in the *Agropyron* genome were indicated by the multiple localization of nine other SSRs, which are summarized in Appendix A.

Finally, 25 out of 159 (15.7%) SSR markers were localized to scaffolds that unambiguously carried genic sequences. Ten of them localized to the specific chromosomes (2P, 3P, 5P, and 6P), and three other markers localized to chromosome 7P. Only 11 markers shared their location on the scaffold with the respective genic sequence on chromosomes 1P, 3P, 4P, 5P, 6P, and 7P (Appendix A).

## 3. Discussion

The present study increases the range of genomic resources for *A. cristatum*, a potential source of important genes for wheat improvement. Chromosome-specific sequence data obtained after the flow sorting of individual chromosomes made it possible to create draft chromosome assemblies and develop new SSR markers. Chromosome-centric genomics has been instrumental for creating and/or improving the draft assemblies of species with large genomes, characterizing genic sequences, creating gene orders, and identifying specific molecular markers (as reviewed by Zwyrtková et al. [58]).

In our study, two chromosome-sorting strategies developed by Šimková et al. [52] and Cápal et al. [51] were used to sequence DNA from individual flow-sorted chromosomes of *Agropyron*. The strategy of Šimková et al. [52] was used to obtain DNA from many copies of chromosomes that were sorted at high purity from specific cytogenetic lines. This approach resulted in draft assemblies that represented a higher proportion of whole chromosome lengths. In comparison, draft assemblies obtained from DNA prepared from single copies of chromosomes, as described by Cápal et al. [51], were less complete, most probably due to higher bias during DNA amplification [51].

Despite the short cumulative length of the draft chromosome assemblies, we were able to identify wild alleles of agronomically important genes highlighting the importance of these genomic resources. The *Agropyron* variants of *HvCslF10* and *HvCslF3* genes on the chromosomes 2P and 3P, respectively, or the gene *TaGT47-13* on 3P could be suitable targets for gene introgression programs aimed at modifying the dietary fiber composition of wheat [59]. The upregulation of transporter genes *SOS1* (a plasma membrane Na^+^/H^+^ transporter) and *HKT1,* which mediates Na distribution between roots and shoots, is a typical response for salt stress in tolerant wheat genotypes [60]. The fact that these genes were found on chromosomes 3P and 7P, respectively, may indicate the role of these chromosomes in the salt tolerance of *A. cristatum*. A careful stress–physiological investigation of wheat–*A. cristatum* addition lines will be needed to confirm this hypothesis. Several gene homologs connected with rust resistance or genes for multidrug resistance were also identified on the chromosome assemblies of *A. cristatum*. In previous studies, SSR and STS molecular markers helped to localize leaf-rust resistance on the 1PS chromosome arm. The 1BL translocation line [17] and other genes of resistance to stripe rust and leaf rust were mapped to chromosome 6PS [61,62], in contrast to in our study, in which the genes homologous to rust resistance were found in draft assemblies of chromosomes 1P, 3P, and 5P. The known positions of genes on individual chromosomes of *Agropyron* or another alien species are necessary to simplify introgression breeding, especially for the creation of better-targeted addition lines in a breeding strategy.

The representation of repetitive DNA sequences identified in draft chromosome assemblies was in concordance with our previous study describing the characterization of the repetitive landscape of the wheatgrass genome from partial genome sequencing [40]. As in other Poaceae species, a large part of the complex genome of *A. cristatum* consists of different types of retrotransposons, with Ty3/gypsy elements being more abundant than Ty1/copia. Variable lineages of Ty3/gypsy elements were also found to be most abundant in closely related species, such as barley [42], rice [63], maize [64], wheat [65], oat [66], rye [67], *Aegilops* [68], fescues, and ryegrasses [68]. However, the genome of wheatgrass contains a relatively large number of different satellite repeats, which enabled the identification of individual chromosomes in situ [40]. Interestingly, none of the satellite repeats were found in centromeric regions. In this study, we identified and further confirmed the presence of *Agrocen*’s LTR element in the centromeric regions of all the wheatgrass chromosomes. Its close phylogenetic relationship to *Cereba*-like elements in Gramineae [54,55,56,57], together with its colocalization with histone CENH3 [69], indicated the role of *Agrocen* retrotransposon in centromere function.

Simple-sequence repeats or microsatellites (SSRs) are highly abundant in plant genomes and widely used in association studies and genetic-diversity studies for the fingerprinting of breeding lines or for the construction of genetic linkage maps [34,37]. In our study, we have identified and developed new *Agropyron*-specific SSR markers that will facilitate the rapid and easy characterization of wheat–*A. cristatum* addition lines. Our chromosome draft assemblies enabled the characterization of 72 chromosome-specific and/or chromosome-arm-specific SSRs, including those for chromosome 7P, expanding our previous study, in which co-dominant COS markers, originally developed for wheat [70,71], were mapped onto six out of seven chromosomes of *Agropyon* [30]. Moreover, we have developed *Agropyron*-specific markers, which have been found distributed in multiple loci on all the *Agropyron* chromosomes but are missing in wheat and, thus, can be used for initial screening for the presence of fragments of *Agropyron* genomes in newly developed wheat–*A. cristatum* introgression lines. In the future, the new SSR markers can be integrated into a marker-assisted selection system to follow the introgressed *Agropyron* chromatin during the transfer into elite cultivars. Our dataset contains many more SSRs identified in silico, which can be used for the development of additional PCR-based chromosome-specific markers or together with other types of markers, e.g., SNPs, for the creation of genetic linkage maps and to enable direct map-based cloning, QTL analysis, or anchoring physical and genetic maps.

In our study, we used newly developed chromosome-specific SSR markers to compare the organization of homologous chromosomes of *Agropyron* and bread wheat. This knowledge is essential for breeders to minimize and/or avoid the possibility of synapsis and chiasma formation in addition lines, which can lead to the elimination of alien (*Agropyron*) chromosomes [72,73]. However, the chromosomes of the tribe Triticeae are highly collinear [74], and a number of chromosomal rearrangements, including inversions and translocations, occurred during their evolution [75]. Newly developed markers confirmed the presence of paracentric inversions on chromosome 4P, which correlated with the previous study of Said et al. [40], who discovered paracentric inversion on 4P using FISH with single-copy cDNA probes. Our findings showed that the structure of chromosome 4P of wheatgrass is collinear with wheat chromosome 4A, which has an inverted structure with respect to chromosomes 4B and 4D [74]. Our results also indicate a short shift in the centromeric region of chromosome 2P and the presence of the non-reciprocal translocation of the pericentromeric region of 1PL to 6PL (Figure 3, Appendix A).

Newly developed SSR markers provided important information about the structure of the chromosomal arms present in wheat–*Agropyron* telosomic addition lines. Our results show that the additional line of chromosomes 6PS and 6PL did not contain whole chromosomal arms and that a part of the centromeric region was missing. This can be explained by the unequal breakage of chromosome 6P and the transfer of incomplete parts into the wheat background or the use of different, structurally heterozygous genotypes for transferring the short and long arms of chromosome 6P into bread wheat. A similar situation was detected for chromosome 2P. Similarly, the amplification of several SSR markers on cytogenetic lines containing chromosome arms 2PS and 2PL most probably indicates that these additional lines consist of overlapping centromeric regions. Our results point to the fact that specific wheat–*Agropyron* addition lines were created from different tetraploid cultivar(s) of wheatgrass and illustrate their genomic variability [76]. Due to the cross-pollinating nature of *Agropyron,* structural variability is also typical for its diploid species [40].

The application of other genomic tools and approaches will be needed to analyze the chromosome structural variation in *Agropyron* more precisely [77]. Although Hi-C and Bionano optical mapping technologies are suitable for the comparison of organisms with highly similar DNA sequences, e.g., different plant cultivars [78,79], the creation of high-density genetic maps or the use of third-generation sequencing technologies, such as Oxford Nanopore, could enable the comparison of the structure of chromosomes between different related species [80,81,82,83,84].

## 4. Materials and Methods

### 4.1. Plant Material and the Isolation of Genomic DNA

Seeds of crested wheatgrass (*Agropyron cristatum*) cv. Parkway (2*n* = 2*x* = 14, PP genome) were provided by Dr. Joseph Robins (ARS Forage and Range Research Laboratory, USDA, Logan, UT, USA). Seeds of common wheat (*Triticum aestivum*) cv. Chinese Spring were provided by Dr. Pierre Sourdille (INRA, Clermont-Ferrand, France). Seeds of wheat (cv. Chinese Spring)–*A. cristatum* disomic chromosome addition lines 1P, 2P, 3P, 4P, 5P, and 6P and wheat (cv. Chinese Spring)–*A. cristatum* telosome addition lines 2PS, 2PL, 4PS, 5PL, 6PS, and 6PL were produced by Chen et al. [21,22] and provided to us by Dr. Adoración Cabrera (University of Córdoba, Córdoba, Spain). Seeds of wheat–*A. cristatum* disomic chromosome 3P addition lines were renamed the 3P + 3PS line in this work, following the results of Said et al. [30]. The wheat (cv. Chinese Spring)–*A. cristatum* 3PS telosome addition line was developed by Said et al. [30]. Seeds of wheat (cv. Chinese Spring)–*A. cristatum* with the Robertsonian translocation line TH4, comprising the long arm of wheat chromosome 1B and the short arm of chromosome 1PS of tetraploid *A. cristatum,* were developed by Ochoa et al. [17] and provided to us by Dr. Adoración Cabrera (University of Córdoba, Córdoba, Spain). Genomic DNA was isolated from the young leaf tissues of plants grown under controlled temperature, light, and humidity regimes using the NucleoSpin^®^ Plant II kit (Macherey-Nagel GmbH & Co. KG, Düren, Germany), following the manufacturer’s instructions.

### 4.2. Chromosome Sorting and the Amplification of DNA

A flow cytometric analysis of suspensions of intact mitotic metaphase chromosomes and chromosome sorting was performed as described by Said et al. [85]. Chromosomes 1P, 2P, 3P, 4P, 5P, and 6P were flow-sorted from wheat–*Agropyron* addition lines, whereas the chromosome 7P, for which the chromosome addition line was not available, was sorted from *A. cristatum* cv. Parkway in a single-chromosome mode [51]. Three independent batches of 100,000 copies of chromosomes 2P, 3P, 5P, and 6P were flow-sorted and checked for identity and contamination using genomic in situ hybridization (GISH). Their DNA was amplified using an Illustra GenomiPhi V2 DNA Amplification Kit (GE Healthcare, Chicago, IL, USA) following Šimková et al. [52]. The amplification products from each chromosome were combined to reduce the potential amplification bias. Chromosomes 1P and 4P were sorted as single copies from respective addition lines, and chromosome 7P was sorted from diploid cultivar Parkway. The DNA of chromosomes 1P, 4P, and 7P was amplified using the single-chromosome amplification procedure according to Cápal et al. [51]. Their identity was assessed using a PCR marker targeting *Agropyron*-specific retroelements [40] prior to sequencing. To identify chromosome 7P, DNA samples were checked using a set of PCR-based markers developed from cDNA sequences, which were previously mapped to chromosome 7P [40]. Five independent and PCR-confirmed amplified products of chromosomes 1P and 7P and six products of chromosome 4P were pooled and used for Illumina sequencing.

### 4.3. Illumina Sequencing

Illumina libraries were prepared from amplified DNA for each chromosome separately using a TruSeq^®^ DNA PCR-Free High Throughput Library Prep Kit (Illumina, San Diego, CA, USA). Chromosomes 2P and 6P were paired-end sequenced on an Illumina MiSeq instrument (Illumina, San Diego, CA, USA) with 300 bp PE reads in several runs to achieve a minimal output of ~20× sequencing depth for each chromosome. Chromosomes 1P, 3P, 4P, 5P, and 7P were sequenced on the HiSeq 2500 instrument in 250 bp PE Rapid Run mode to achieve a minimal output of ~20× sequencing depth for each chromosome. The sequence reads were deposited in the Sequence Read Archive (BioProject ID: PRJNA801633; accessions SAMN25358569, SAMN25358571, SAMN25358572, SAMN25358575, SAMN25358577, SAMN25358579, SAMN25358581, SAMN25358582, and SAMN25358585). Assembled contigs and scaffolds specific for individual wheatgrass chromosomes are publicly available in the Dryad Digital Repository: https://doi.org/10.5061/dryad.gqnk98spv (accessed on 12 February 2022).

### 4.4. De Novo Assembly, Identification, and Verification of Chromosome-Specific SSR Markers

Raw datasets obtained after the Illumina sequencing of the seven *A. cristatum* chromosomes were trimmed to the appropriate length and quality (-q 20 -p 90) using Trimmomatic [86] and assembled using a Ray de novo assembler [87] with the following settings: a k-mer length of 75, a minimum contig length of 100 bp.

The identification of SSR markers was carried out using the microsatellite identification tool (MISA) [88], and the primers for the PCR amplification of the identified SSR markers were designed using Primer3 [89]. With the aim of increasing the probability of identifying chromosome-specific markers, *Agropyron* chromosomal scaffolds containing unique SSRs were mapped onto the wheat genome sequence assembly RefSeq v1.0 [46] using Gmap [90]. Based on the collinearity with bread wheat, a set of SSR markers that mapped to unique sites on the wheat D-subgenome were selected and further verified using PCR.

The PCR contained 20 ng of template DNA, a 1.25 µM mix of forward and reverse primers, 200 µM dNTPs, 1 mM MgCl_2_, and 2U/100 µL of the final reaction’s volume Taq polymerase with 1× Taq buffer (New England Biolabs, Ipswich, MA, USA) and was performed under the following conditions: 94 °C for 5 min, 35× (94 °C for 50 s, an appropriate annealing temperature for each primer pair for 50 s, and 72 °C for 50 s), and 72 °C for 5 min. Genomic DNA was isolated from wheat–*Agropyron* telosome addition and translocation lines, diploid *A. cristatum* cv. Parkway, and *T. aestivum* cv. Chinese Spring for use as the template DNA. The best annealing conditions were chosen after gradient PCR. The PCR products were analyzed by electrophoresis on a 2% agarose gel.

### 4.5. Annotation of Chromosome-Specific Sequences

The annotation of *A. cristatum* scaffolds was performed using the MAKER-P pipeline [53]. The use of the pipeline comprised two steps. The first step involved masking the genome using RepeatMasker [91] and the de novo reconstruction of DNA repeats of *A. cristatum* (https://olomouc.ueb.cas.cz/en/content/dna-repeats (accessed on 12 February 2022)); [40]. The resulting annotation of repeated elements was then added as an input GFF file for the second step. Genic sequences were identified using the gene-prediction tools AUGUSTUS [92] and FGENESH (http://www.softberry.com (accessed on 12 February 2022)) with the wheat and monocot models, respectively. The genic sequence prediction was supported by protein homologies with *Arabidopsis thaliana*, *Brachypodium distachyon*, *Oryza sativa*, *Triticum aestivum*, *Sorghum bicolor*, *Zea mays*, and *Aegilops tauschii*, the sequences of which were retrieved from Ensembl Plants release 45 [93], and *Hordeum vulgare* proteins from the second assembly version [94]. The resulting potentially expressed sequences and proteins were then filtered to keep only the longest splicing variant for each genic sequence.

Following the MAKER-P pipeline, the functional annotation of the genic sequences was performed using InterProScan version 5.36-75.0 [95] and BLAST [96] results with the UniProt/SwissProt version from 2019/08 [97]. The clustering of genic sequences into functional families was conducted using SiLiX [98].

### 4.6. Immunostaining of Interphase Nuclei

BLASTN homology searches and further phylogenetic analysis of reverse transcriptase domains of representatives of Ty3/gypsy elements [57] and repetitive DNA sequences characterized in our previous study [40] were performed according to Novák et al. [99]. Multiple sequence alignment of reverse transcriptase domains was done with MAFFT v7.029 (–globalpair –maxiterate 1000) [100]. Phylogenetic trees were constructed with PhyML 3.0 [101] implemented in SeaView v5.0.2 [102]. The approximate likelihood ratio test [103] was performed to assess branch support. The phylogram was depicted in FigTree (http://tree.bio.ed.ac.uk/software/figtree/ (accessed on 12 February 2022)). The *Agropyron*-specific DNA element, which was identified as the closest relative to the *Cereba* element, was selected and used for the creation of the cytogenetic probe, which was co-localized with the anti-CENH3 antibody.

The cell nuclei of *A. cristatum* cv. Parkway were flow-sorted into PRINS buffer supplemented with 2.5% sucrose [104]. Microscopic slides with the sorted nuclei were stored at room temperature overnight. Fluorescence in situ hybridization (FISH) and immunostaining were carried out according to [68], with minor modifications: probes derived from the reverse transcriptase (RT) domain of the centromeric element were labeled with biotin-16-dUTP (Roche Applied Science) using PCR with specific primers (forward: 5′-GATGGTACGTCGCGTATGTG-3′; reverse: 5′-CGTCCAATGAAAGCACGTAA-3′); the primary antibody anti-CENH3 [105] was diluted 1:125, and the secondary antibodies (anti-rabbit antibodies STAR 635P, ST635-1002-500UG, Abberior, Göttingen, Germany) were diluted 1:200. The slides were examined with a confocal microscope (Leica Microsystems, Wetzlar, Germany) with appropriate laser lines, and images were prepared with Leica Application Suite X (LAS-X) software version 3.5.5 with the Leica Lightning module (Leica, Buffalo Grove, IL, USA). The pictures were deconvoluted with the Leica Lightning module based on the refractive index of the mounting medium. The final images were arranged in Adobe Photoshop 12.0 (Adobe Systems Corporation, San Jose, CA, USA).

## 5. Conclusions

Our work has created new genomic resources and increased the knowledge about the genome/chromosome structure of *A. cristatum*. Chromosome draft assemblies were created, and regions homologous to agronomically important genes were identified. The classification of repetitive DNA sequences revealed the centromere-specific LTR element *Agrocen*, whose colocalization with histone CENH3 indicates its role in centromere function. Simple-sequence repeats were identified in chromosome draft assemblies, and 72 new SSR markers specific to all seven chromosomes or chromosome arms of *Agropyron* were developed and used to compare the homologous chromosome structures between *Agropyron* and bread wheat. We demonstrate the potential of newly developed SSR markers for the rapid characterization of wheat–*A. cristatum* introgression lines to accelerate introgression breeding programs.

## Figures and Tables

**Figure 1 ijms-23-03191-f001:**
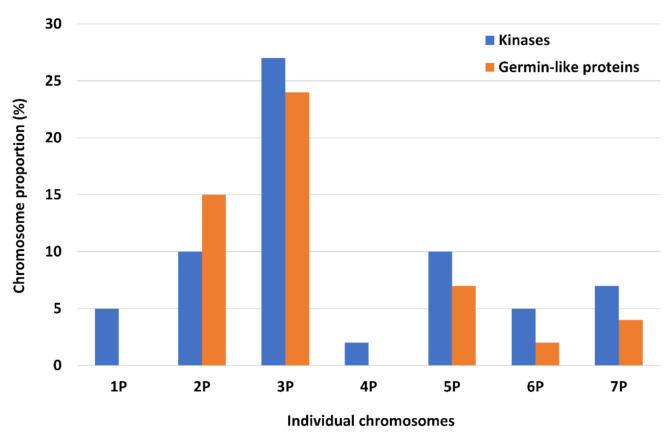
Chromosomal distribution of the two most represented gene functional families observed in the draft sequencing of *A. cristatum*. *X*-axis: individual *Agropyron* chromosomes; *Y*-axis: chromosome proportion (%).

**Figure 2 ijms-23-03191-f002:**
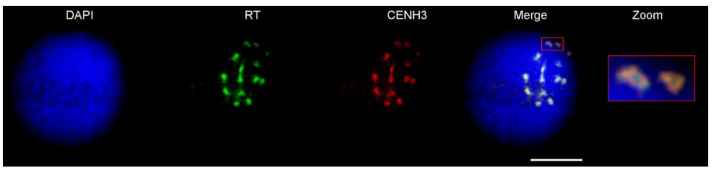
Colocalization of CENH3 with the *Agrocen* element in *A. cristatum* cv. Parkway. A combination of the immunolocalization of the histone H3 variant CENH3 (red) and FISH on interphase nuclei with probes for the reverse transcriptase (RT) domain (green). Nuclei were counterstained with DAPI (blue). Bar corresponds to 10 µm.

**Figure 3 ijms-23-03191-f003:**
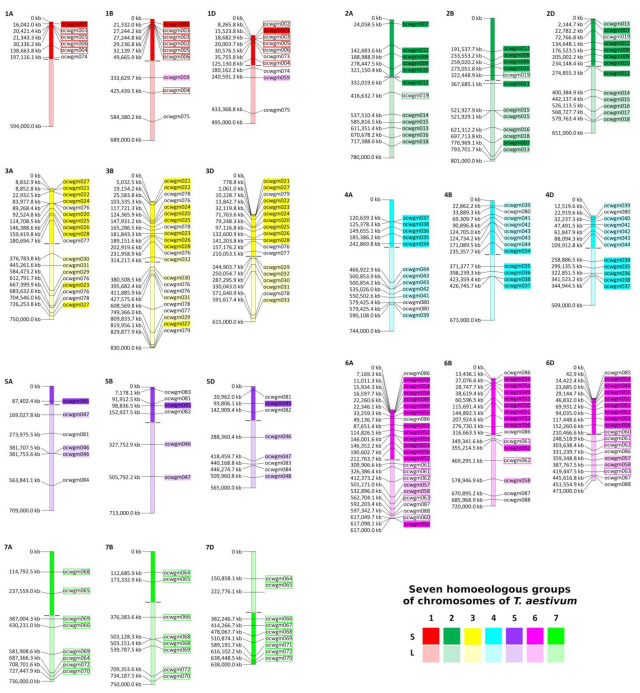
Position of the new *Agropyron*-specific SSR markers on collinear chromosomes of *T. aestivum*. Seven homoeologous groups of wheat chromosomes are differentiated by colors (red, green, yellow, turquoise, purple, pink and light green). Short chromosomal arms (S) are dark in color, and long chromosomal arms (L) are light in color. Chromosome-specific SSR markers are marked by frames. Chromosome-arm-specific SSR markers are marked by frames filled up by colors of the corresponding chromosome arms. Any specific highlighting is not used for SSR markers localized in multiple positions in the genome of *A. cristatum*. Arrows depict positions of centromeres, as observed by Appels et al. [46].

**Figure 4 ijms-23-03191-f004:**
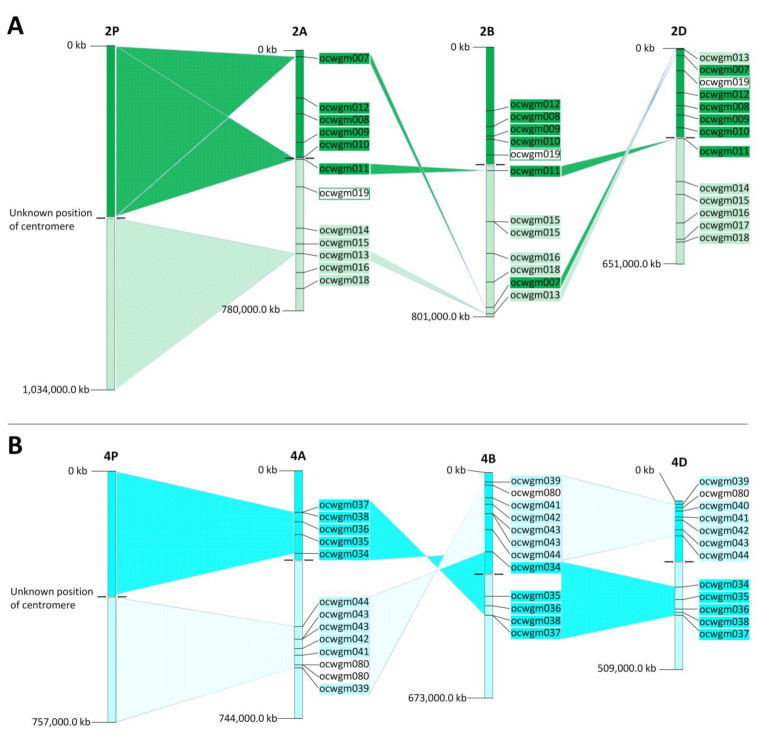
Rearrangements specific for chromosome group 2 between *A. cristatum* and common wheat [46] (**A**), and rearrangements specific for chromosome group 4 between *A. cristatum* and common wheat [46] (**B**), as revealed by newly developed SSR markers.

**Table 1 ijms-23-03191-t001:** Sequencing of *A. cristatum* chromosomes.

Chromosome *									
	Purity (%)	Length (Mb) **	RayAssembly (bp)	N50 (bp)	Coverage (%) ***	Number of SSRs	Maker ^&^	Maker ^&&^	Genes ^&&&^
1P	100	870	99,700,722	5724	11.6	5949	3322	3307	758
2P	90	1034	342,722,238	3774	33.2	19,604	7759	7700	1093
3P	98	791	443,871,941	5973	56.1	32,624	11,045	10,976	1388
4P	100	757	32,166,648	8434	4.3	1136	1472	1470	749
5P	98	969	463,809,210	8175	47.9	34,401	11,769	11,694	1564
6P	81	974	258,122,996	2218	26.5	30,932	6817	6756	867
7P	100	958	147,241,484	4886	15.4	9624	2942	2911	499

* Single copies of chromosomes 1P, 4P, and 7P were flow-sorted. ** Based on Said et al. [40]. *** Chromosomal coverage of the assemblies. ^&^ Potentially expressed sequences. ^&&^ Genic sequences. ^&&&^ Genic sequences with known function.

**Table 2 ijms-23-03191-t002:** The most accurate locations of specific SSR markers established with *Agropyron* addition lines.

Chromosome Addition Line	Chromosome-Short-Arm Addition Line	Chromosome-Long-Arm Addition Line
1P	1PS + 1BL	No line
ocwgm001	ocwgm001	
ocwgm002	-	
ocwgm003	-	
ocwgm004	-	
ocwgm005	-	
ocwgm006	-	
2P	2PS	2PL
ocwgm007	ocwgm007	-
ocwgm008	ocwgm008	-
ocwgm009	ocwgm009	-
ocwgm010	ocwgm010	-
ocwgm011	ocwgm011	-
ocwgm012	ocwgm012	-
ocwgm013	-	ocwgm013
ocwgm014	-	ocwgm014
ocwgm015	-	ocwgm015
ocwgm016	-	ocwgm016
ocwgm017	-	ocwgm017
ocwgm018	-	ocwgm018
ocwgm019	ocwgm019	ocwgm019
3P + 3PS	3PS	No line
ocwgm020	ocwgm020	
ocwgm021	ocwgm021	
ocwgm022	ocwgm022	
ocwgm023	ocwgm023	
ocwgm024	ocwgm024	
ocwgm025	ocwgm025	
ocwgm026	ocwgm026	
ocwgm027	ocwgm027	
ocwgm028	ocwgm028	
ocwgm029	-	
ocwgm030	-	
ocwgm031	-	
ocwgm032	-	
ocwgm033	-	
4P	4PS	No line
ocwgm034	ocwgm034	
ocwgm035	ocwgm035	
ocwgm036	ocwgm036	
ocwgm037	ocwgm037	
ocwgm038	ocwgm038	
ocwgm039	-	
ocwgm040	-	
ocwgm041	-	
ocwgm042	-	
ocwgm043	-	
ocwgm044	-	
5P	No line	5PL
ocwgm045		-
ocwgm046		ocwgm046
ocwgm047		ocwgm047
ocwgm048		ocwgm048
6P	6PS	6PL
ocwgm049	ocwgm049	-
ocwgm050	ocwgm050	-
ocwgm051	ocwgm051	-
ocwgm052	ocwgm052	-
ocwgm053	ocwgm053	-
ocwgm054	ocwgm054	-
ocwgm055	ocwgm055	-
ocwgm056	ocwgm056	-
ocwgm057	-	ocwgm057
ocwgm058	-	ocwgm058
-	-	ocwgm059
ocwgm060	-	-
ocwgm061	-	-
ocwgm062	-	-
ocwgm063	-	-
7P *	No line	No line
ocwgm064		
ocwgm065		
ocwgm066		
ocwgm067		
ocwgm068		
ocwgm069		
ocwgm070		
ocwgm071		
ocwgm072		

* Absent on other addition lines; present on gDNA.

## Data Availability

The sequence reads were deposited in the Sequence Read Archive (BioProject ID: PRJNA801633; accessions SAMN25358569, SAMN25358571, SAMN25358572, SAMN25358575, SAMN25358577, SAMN25358579, SAMN25358581, SAMN25358582, and SAMN25358585). Assembled contigs and scaffolds specific for individual wheatgrass chromosomes are publicly available from the Dryad Digital Repository: https://doi.org/10.5061/dryad.gqnk98spv (accessed on 12 February 2022).

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
