# Peer review of "Draft Sequencing Crested Wheatgrass Chromosomes Identified Evolutionary Structural Changes and Genes and Facilitated the Development of SSR Markers"

_ijms, 2022, doi:10.3390/ijms23063191_

Round 1
Reviewer 1 Report
Zwyrtková et al. perform a clever method to identify and map the chromosomal location of genes and repeats in non-model organisms with large and poorly characterized genomes, and are able to develop new SSR markers with this information. Their results in Agropyron cristatum, a wild relative of wheat and barley, are potentially useful for their improvement by introgression breeding.
I only have some minor comments for the authors.
Line 41. Please, make clear that Triticum aestivum is what we know as common wheat.
Line 41. How long ago did T. aestivum and A. cristatum diverged? Can A. cristatum be introgressed into barley? (Barley is mentioned in the abstract but not further in the text)
Table 1. Maybe simplify column names so they are not so long, and add further description into the table caption if needed.
Line 174. “A phylogenetic analysis of the RT domain of Agrocen LTR elements confirmed their close relationship to centromeric retrotransposons” I cannot find data supporting this statement, and neither the description of this analysis in Methods. Also, specify to which species.
Line 201. The meaning of “polymorphic pattern” is unclear.
Line 205. “However, a possibility of their absence on 7P could not be excluded” This statement is confusing, try to rephrase.
Line 252. Please show also the rearrangements inferred from the three markers in 2P
Author Response
We appreciate the valuable comments and advice on how to improve the manuscript.
1) Line 41. Please, make clear that Triticum aestivum is what we know as common wheat.
Response: We have modified the text accordingly.
2) Line 41. How long ago did T. aestivum and A. cristatum diverged? Can A. cristatum be introgressed into barley? (Barley is mentioned in the abstract but not further in the text)
Response: We apologize for the mistake in the Abstract, where barley was mentioned. Agropyron cristatum is used to create introgression lines of wheat. The Abstract was corrected.
3) Table 1. Maybe simplify column names so they are not so long, and add further description into the table caption if needed.
Response: We appreciate the suggestions. The column names were shortened.
4) Line 174. “A phylogenetic analysis of the RT domain of Agrocen LTR elements confirmed their close relationship to centromeric retrotransposons” I cannot find data supporting this statement, and neither the description of this analysis in Methods. Also, specify to which species.
Response: We apologize for omitting this information. We have included a description of phylogenetic analysis of Ty3-gypsy elements in the Methods section. We have also included a phylogram in the Supplementary Figure 2 and modified the legend to the Figure accordingly.
5) Line 201. The meaning of “polymorphic pattern” is unclear.
Response: We have modified the text to clarify the meaning.
6) Line 205. “However, a possibility of their absence on 7P could not be excluded” This statement is confusing, try to rephrase.
Response: We have revised the main text to clarify this statement.
7) Line 252. Please show also the rearrangements inferred from the three markers in 2P
Response: We have added a scheme of 2P rearrangement in Figure 2. We have revised a legend to the Figure 2 accordingly.
Reviewer 2 Report
Agropyron cristatum, is an important source of genes and alleles for genetic improvement of wheat. This study aims to identify molecular markers to provide insights into the genome organization of Agropyron cristatum and facilitate the introgression breeding of wheat.
The genic sequences homologous to genes conferring plant disease resistance and involved in plant tolerance to biotic and abiotic stress were identified. However, in my opinion it is not clear how the genic sequences, the repetitive DNA, the SSR markers identified in this work can affect wheat improvement. What stress do they give resistance to and why? This part should be clarified in the discussions.
Author Response
We thank the reviewer for valuable comments and suggestions on how to improve the manuscript. We have included additional information in the Discussion to support our statement and to answer the Reviewer’s questions.